# Development and Optimization of Chromatographic Conditions for the Determination of Selected B Vitamins in Pharmaceutical Products

**Joanna Żandarek** [1,2], **Żaneta Binert-Kusztal** [1], **Małgorzata Starek** [1] and **Monika Dąbrowska** [1,*]

1 Department of Inorganic and Analytical Chemistry, Faculty of Pharmacy, Jagiellonian University Medical College, 9 Medyczna St, 30-688 Kraków, Poland
2 Doctoral School of Medical and Health Sciences, Jagiellonian University Medical College, 16 Łazarza St, 31-530 Kraków, Poland
* Correspondence: monika.1.dabrowska@uj.edu.pl

**Abstract:** Vitamins are a unit of organic chemical substances that are essential for the adequate working of the human body. Water-soluble B vitamins are involved in the regulation of many metabolic and regulatory processes. Due to the inability to synthesize endogenously, they must be supplied to the body with edibles or in the form of supplementation as drugs or dietary supplements. Maintaining the correct level of vitamins is extremely important in the treatment of various diseases. In the presented work, the qualitative and quantitative procedure of the assay of vitamins B1, B2, B5, B6 and B12 in pharmaceutical products by chromatographic technique coupled with densitometric detection was developed, optimized and validated. During the optimization process, TLC Silica gel 60 F254 plates were chosen as a suitable stationary, and the mixture consisted of chloroform: ethanol: water: glacial acetic acid (2:8:2:0.5 *v/v/v/v*) as a mobile phase. Densitometric detection was conducted at a maximum absorbance λ = 254 nm for vitamins B1, B2, B6 and B12 and λ = 550 nm for vitamin B5 (after dyeing with ninhydrin). In the next step, the developed procedure was validated in accordance with the ICH guidelines. The recorded correlation coefficients obtained in all tested concentrations of B vitamins, ranging from 0.9947 to 0.9996, confirmed good linearity. The method is characterized by good precision, RSD data ranging from 0.62 to 1.52% for direct precision and from 0.84 to 1.4% for intermediate precision. Accuracy was proven by a recovery test at three concentration levels, with values close to 100% with RSD less than 1%. The calculated LOD and LOQ data for all tested vitamins B1, B2, B5, B6 and B12 were belove 1 μg/spot. The developed method was then used to quantitatively and qualitatively assess the content of B vitamins in medicinal products and dietary supplements with satisfactory results.

**Keywords:** vitamin B; TLC-densitometry; validation method

## 1. Introduction

Vitamins are a group of organic chemical compounds necessary for the proper functioning of the human body. They are not synthesized endogenously, so there is a need to supply them with food. Vitamins are involved in the metabolism, differentiation and growth of cells. Insufficient intake of these compounds, reduced absorption in the intestines, disease states and genetic disorders are potential causes of deficiencies that have a negative impact on human health. Vitamins can be divided into two groups: fat soluble (A, D, E, K) and water soluble, which include the following: vitamin C and a group of eight vitamins B (B1—thiamin, B2—riboflavin, B3—niacin (nicotinamide), B5—pantothenic acid, B6—pyridoxine, B7—biotin (vitamin H), B9—folic acid, B12—cobalamin) [1]. Knowledge about vitamins is constantly expanding. More accurate methods of their detection and determination using chemical, physical, physicochemical or biological methods are also

being developed. Such analyses are carried out in diagnostics (testing the level of vitamins in blood and urine), as well as in quality control of pharmaceutical products. Most studies use chromatographic, spectrophotometric, spectrofluorimetric and electroanalytical techniques [2].

The determination of water-soluble B vitamins (Figure 1) is quite difficult due to their chemical instability. Many analytical methods turn out to be time consuming or inaccurate. The most commonly used technique is reverse-phase high-performance liquid chromatography (RP-HPLC), which is characterized by high selectivity and sensitivity [2]. Vitamin B1 is a component of many enzymes, including $\alpha$-ketoglutarate dehydrogenase necessary for the reaction in the Krebs cycle and dehydrogenase pyruvate, which plays an important role in carbohydrates metabolism. In addition, it is a coenzyme in the pentose phosphate pathway necessary for the synthesis of nucleic acids, steroids and fatty acids [1]. It plays a neuromodulatory role, affecting nerve conduction, the proper functioning of nerve cells and the brain and by reacting with free radicals inhibiting the unfavorable lipid peroxidation in liver microsomes [2]. Long-term administration of vitamin B may be effective in arthritis because it has analgesic, anti-inflammatory effects and lowers the level of cytokines TNF-$\alpha$ and IL-1$\beta$ in the blood [3]. Thiamine deficiency leads to beriberi disease or Wernicke–Korsakoff syndrome, and has been linked to Alzheimer's disease, dementia, depression and diabetes [1]. Among the thiamine determination methods, HPLC dominates [4–6]. It has been used for vitamin B1 analysis in the blood or for the time-dependent monitoring of thiamine concentration in human urine [4,5,7]. Modifications to this technique are sometimes used such as hydrophilic interaction chromatography (HILIC), which allows the separation of polar and hydrophilic compounds using a hydrophilic stationary phase and a water–organic mobile phase with a high content of organic solvent (60–95%) [7]. Thiamine can also be determined spectrophotometrically, e.g., using its oxidation in reaction with potassium (V) iodate [8]. Spectrophotometric methods are fast and simple, but they carry the risk of interference, especially in the case of biological samples. Less popular are techniques using microorganisms or enzymes. By monitoring the growth of microorganisms for which thiamin is essential, its levels can be determined. However, these methods are associated with many problems, including the following: the availability of specific organisms, the effect of other sample components on microorganisms and the prolonged time of their growth. Vitamin B1 is a cofactor for many enzymes. The percentage increase in their activity is used to determine the thiamine content. Unfortunately, this method also has disadvantages; tests do not provide direct information about the concentration of the vitamin, the affinity of the apoenzyme may be reduced in some people and the increase in enzyme activity may be caused by the presence of another cofactor [1].

Vitamin B2 is an essential ingredient for the proper functioning, growth and development of cells in all aerobic life forms. Riboflavin is a precursor to flavin mononucleotide (FMN), which is then metabolized to flavin dinucleotide (FAD). These compounds act as coenzymes for many oxidases and reductases in metabolic processes [2]. Vitamin B2 is involved in the activation of the immune system to fight cancer and enhances the effect of anti-cancer drugs. Research shows that supplementation with riboflavin can reduce the frequency of migraine headaches [9]. Vitamin B2 participates in the synthesis of red blood cells, helps maintain the proper functioning of the digestive system and is involved in the metabolism of homocysteine [10]. Vitamin B2 deficiencies can lead to vision disorders, digestive disorders, dermatitis and abnormalities in the production of red blood cells [2]. Many procedures have been developed for the vitamin B2 determination, mainly based on HPLC with spectrometric detection or coupled with tandem mass spectrometry (MS/MS), which is characterized by high selectivity [11,12]. Vitamin B2 was determined in human urine [13], and, using fluorometric detection, both in urine and blood serum [14]. Riboflavin was also determined by UV spectrophotometry in aqueous solutions. These methods are often used in quality control and for monitoring the production process in industry [15]. The available literature also includes enzymatic, microbiological and electrochemical methods [16]. In the case of biological samples, the use of voltammetry and a carbon paste

electrode has proven to be effective, resulting in high sensitivity and selectivity as well as a low detection limit [17].

**Figure 1.** Chemical structure of the tested B vitamins.

Vitamin B5 can exist in several forms. The most common in pharmaceuticals is D-pantothenic acid. In turn, the most active form of this vitamin is panthein, which contains a sulfhydryl group necessary for the operation of coenzyme A (Co-A), which acts as a cofactor in many metabolic transformations of carbohydrates, proteins and lipids, and in the synthesis of neurotransmitters and hormones. The liquid form of vitamin B5 is dexpanthenol, used externally to accelerate wound healing and treat acne. Vitamin B5 helps in the proper functioning of the digestive, nervous and circulatory systems. Its deficiency can cause weight loss, fatigue, skin inflammation, dyslipidemia, neuropathy and adrenal dysfunction [18]. The determination of vitamin B5 was performed using the microbiological method, but it is a time-consuming, imprecise and non-specific method. On the other hand, radioimmunoassays and indirect immunoassays have drawbacks related to their practical implementation (e.g., the need to use radioisotopes) [19]. The most commonly used method for the determination of B5, both in the form of pantothenic acid [20,21] and in the presence of other vitamins [22,23], is liquid chromatography.

The term vitamin B6 includes six active compounds with a similar structure and physiological effect: pyridoxine (PN), pyridoxamine (PM), pyridoxal (PL) and their phosphorylated derivatives. Vitamin B6 is involved in the metabolism of proteins, carbohydrates and lipids, and affects the proper functioning of the immune and endocrine systems. It is also used to treat hypertension and reduces the neurotoxicity of cisplatin and fluoropyrimidine without weakening the effect of anticancer drugs [24]. Researchers have shown that pyridoxine supplementation in migraine sufferers reduces the severity of headaches and the duration of attacks [25]. Vitamin B6 deficiency leads to impaired growth of the body and inflammation of the skin, and side effects occur in the nervous system, resulting in coordination disorders, hyperactivity or convulsions [24]. Assays of vitamin B6, such as other B vitamins, are most often performed by HPLC. Ion exchange chromatography or reverse-phase ion-pair chromatography is sometimes used due to the dependence of the ionic nature of the vitamin on pH [26–28]. The choice of mobile phase composition depends on the sample extraction method and the desired pH, which is the most optimal

for separation. The HPLC technique has been used to determine vitamin B6 in food, human blood, cerebrospinal fluid or pharmaceutical preparations [29,30]. The spectrophotometric method (at 480 nm) was used to determine pyridoxine in a pharmaceutical preparation, based on the coupling reaction of the vitamin with diazotized p-nitroaniline in an alkaline solution in the presence of CTAB (cetyltrimethylammonium bromide) [31]. An alternative technique for the quantification of vitamin B6 in multivitamin supplements and food is voltammetry, which uses a screen of disposable electrodes [32]. In enzymatic methods, the dependence of enzyme activity on the concentration of its specific coenzyme is used. The reaction metabolite was labeled with tritium; after determining its radioactivity, the concentration of PLP was calculated [24].

Vitamin B12 is another substance necessary for the proper functioning of the human body. Its biologically active form is mainly methylcobalamin, which is an important coenzyme in many biochemical reactions such as the following: fatty acid oxidation, branched-chain amino acid breakdown, folic acid metabolism in DNA synthesis and protein methylation processes in the nervous system. In preparations, it is most often in the form of cyanocobalamin, which is stable and easily turns into active forms after consuming. Cobalamin deficiency is most common in the elderly, vegetarians and in the course of various diseases, including Crohn's disease and celiac disease. Taking certain medications including proton pump inhibitors, acetylsalicylic acid and metformin may reduce the absorption of vitamin B12 [33]. The most common method for the determination of vitamin B12 is HPLC [34–36]. Since cobalamin contains a cobalt atom, spectroscopy can be used, and the assays based on the formation of complexes B12 with various compounds are detectable at certain wavelengths [37]. High sensitivity and specificity show radioimmunoassays based on the reaction of the antibody and radiolabeled antigen (vitamin B12). They allow the detection of very low concentrations of substances, e.g., in the blood [38]. Other techniques used for determinations, found in the literature, include the following: electroluminescence, capillary electrophoresis and inductively coupled plasma mass spectrometry (ICP-MS) [39].

In the available literature, methods for the determination of vitamins B by thin-layer chromatography (TLC) are less common. An example would be analysis tablets containing vitamins B1, B6 and B12 using a mobile phase composed of the following: chloroform: ethanol: water: acetic acid (2:8:2:0.5 *v/v/v/v*). For the experiment, silica gel 60 F plates with dimensions of 20 × 20 cm and a layer thickness of 0.2 mm were used. The obtained results of limits of detection (LOD) and limits of quantification (LOQ) reached the followed values: 0.05 and 0.1 µg/spot for vitamin B1 and B12 (with a linearity range 0.1–1.5 µg/spot), while for vitamin B6, the results were 0.30 and 0.5 µg/spot (linearity range 0.5–3.5 µg/spot) [40]. Panahi et al. described the HPTLC method for determination of vitamins B1, B2, B6 and B12. As the mobile phase, a mixture of a composition was used as follows: ethanol: chloroform: acetonitrile: toluene: ammonia: water (7:4:4.5:0.5:1:1 *v/v/v/v/v/v*) and as stationary phase, silica gel G60 F254 plates, washed with methanol and chloroform (1:1, *v/v*) before use and 24 h air dried. Densitometric detection was conducted at 252 nm. The LOD and LOQ values were obtained as follows: for B1 42.52 and 141.72, for B2 12.72 and 42.41, for B6 30.09 and 100.31, for B12 5.45 and 11.50 ng [41].

Due to the advantages of the TLC with densitometric detection, which include a simple analysis process with a short development time, quick separation of most compounds, easy visualization of separated components of a complex matrix, as well as the fact that the separation process is faster and the selectivity towards compounds is high (even small differences in the composition are enough to clearly separate), the ability to quickly assess the purity of a given sample, the ability to simultaneous determine components during one analysis, as well as lower cost, make it an alternative to the HPLC technique, commonly used in this type of research [42].

The aim of the presented work was to develop a method for the qualitative and quantitative determination of vitamins B1, B2, B5, B6 and B12 in free form and in selected pharmaceutical products by the thin-layer chromatography with densitometric detection.

The detailed research plan included the following: (i) development and optimization of the conditions of chromatographic separation and determination of mentioned vitamins B; (ii) validation of the method in accordance with ICH guidelines; (iii) application of the developed method to analyze the content of individual vitamins in pharmaceutical preparations available on the Polish market. In the presented work, we proposed a quick, simple method of qualitative and quantitative analysis of B vitamins, which can be used for routine quality control of single- or multi-component pharmaceuticals and dietary supplements.

## 2. Materials and Methods

### 2.1. Standard Substance

Vitamin B1: thiamine hydrochloride (SLBW7025), vitamin B2: riboflavin (WXBC7130 V), vitamin B5: D-pantothenic acid hemi calcium salt (BCBV5786), vitamin B6: pyridoxine (STBH1389), vitamin B12 (MKCF8778) were purchased from Sigma-Aldrich (St. Louis, MA, USA).

### 2.2. Pharmaceutical OTC (Over-the-Counter) Preparations Tested

All preparations were purchased at local pharmacies.

Products containing analyzed vitamins were tested as follows:

B1—Product A (50 mg/tbl.), Product C (3 mg/tbl.), Product D (35 mg/tbl.), Product F (2.4 mg/tbl.), Product G (B1—3 mg/tbl.); B2—Product E (5 mg/tbl.), Product G (5 mg/tbl.), Product H (3 mg/tbl.); B6—Product B (B6—50 mg/tbl.); Product E (4.1 mg/tbl.), Product G (5 mg/tbl.); B12—Product E (700 µg/tbl.); B5—Product I (500 mg/caps.).

### 2.3. Chemicals and Apparatus

Ethanol, chloroform and toluene were purchased from POCH (Gliwice, Poland), glacial acetic acid from Stanlab, (Lublin, Poland), ammonia from CHEMPUR (Piekary Śląskie, Poland), 1% β-cyclodextrin water solution (ex tempore) from Sigma-Aldrich (Steinheim, Germany), 1% ninhydrin etanolic solution (ex tempore) from Merck (Darmstadt, Germany), acetonitrile and water from Witko (Łódź, Poland). All chemicals were an analytical grade.

Densitometer TLC Scanner 3 with Cat4 software (Camag, Muttenz, Switzerland), Linomat V (Muttenz, Camag, Switzerland), analytical balance WPA 120/C/1 (Radom, Radwag, Poland), micro-syringe (Hamilton, USA) and UV lamp with a TLC cabin 254/366 nm (Camag, Muttenz, Switzerland), laboratory dryer Ecocell (BMT, Brno, Czech Republic), chromatografic chamber in size 18 × 16 × 8 cm (Sigma-Aldrich, Laramie, WY, USA) were used.

Chromatographic plates in size 20×20 cm such as Aluminiumoxid 60 $F_{254}$ neutral (Type E, No. 5550), Polyamid 11 $F_{254}$ (No. 5555), Kieselgur $F_{254}$ (No. 5568), TLC Silica gel 60 $F_{254}$ (No. 1.05559.0001), HPTLC Silica gel 60 $F_{254}$ (No. 1.05548.0001), HPTLC Cellulose (No. 1.16092.0001), TLC silica gel 60 RP-18 $F_{254}$S (No. 1.05559.0001), TLC Cellulose (No. 1.05574.0001) were purchased from Merck (Darmstadt, Germany).

### 2.4. Standard and Sample Solutions

Calculated masses of standard substances were dissolved in specific volumes of methanol, obtaining solutions with concentrations of 0.01%, 0.1% or 1% for individual vitamins. The calculated amount of the powdered tablet mass of each test preparation was dissolved in the specified volumes of methanol to obtain solutions containing vitamins of the desired concentrations. Any precipitates were centrifuged (3000 rpm, 7 min) and filtered. Three replicates were performed for each assay.

### 2.5. Optimization of Determination Conditions

In order to select the optimal conditions for the separation and determination of tested B vitamins, 10 mm wide spots of the standard solutions and their mixtures were placed on all tested types of plates. Four different mobile phases were tested with the

composition of chloroform: ethanol: water: glacial acetic acid (2:8:2:0.5 *v/v/v/v)*, ethanol: chloroform: acetonitrile: toluene: ammonia: water (7:4:4.5:0.5:1:1 *v/v/v/v/v/v)*, n-hexane: ethyl acetate (9:1 *v/v),* and 1% β-cyclodextrin solution: methanol (15:1 *v/v).* Chromatograms were developed and, after drying, viewed under a UV lamp and densitometrically scanned. In addition, different developing lengths of the chromatographic plates (95 and 145 mm) were tested. During the course of conducted research, it was found that vitamin B5 should be stained with 1% ninhydrin ethanolic solution. After development, the chromatograms were allowed to dry for 24 h and then immersed in 1% ninhydrin ethanolic solution () in a petri dish. Vitamin B5 turned violet and was visible in the daylight. The drying time in an oven was also optimized. The following heating periods were tested: 10, 15, 20 min. Compared to the peak areas, it was found that the extension of the heating time had no effect on the results. Ultimately, the plates were placed in an oven for about 10 min at a temperature of 85 °C. After approx. 10 min (at room temperature), the plates were densitometrically scanned.

### 2.6. Chromatographic Conditions

Taking into account the results described in the previous chapters, optimal conditions for the determination of vitamins in pharmaceutical preparations using the thin-layer chromatography technique were selected. Standard solutions and solutions of appropriate preparations with the same concentration of vitamins in the amount of 10 μL were applied to TLC plates Silica gel 60F$_{254}$ using Linomat V sample applicator (CAMAG, Muttenz, Switzerland), with a rate of 300 nL/s. The 10 mm wide stripes were placed 10 mm apart and 10 mm from the edge of the plate. The chromatographic separation was performed on 95 mm path with a mobile phase consisting of chloroform: ethanol: water: glacial acetic acid (2:8:2:0.5 *v/v/v/v*) in the chromatographic chamber in size 18 × 16 × 8 cm (Sigma-Aldrich, Laramie, WY, USA), first saturated (15 min) at room temperature with the selected mobile phase. After drying at room temperature (24 h), the plates were scanned with a densitometer (TLC Scanner 3 with winCATS 4 software v.1.44) at 254 nm with slit dimensions 4.00 × 0.45 mm. In contrast, for vitamin B5 assays, plates were stained with an ethanolic solution of ninhydrin (1%) and, after drying in an oven (10 min, at 85 °C), densitometically analyzed at 550 nm.

The values of the R$_F$ coefficients (retardation factor) and the absorption spectra for each vitamin were used for the qualitative analysis.

### 2.7. Method Validation

Method validation is the process of assessing the suitability of an analytical method for a specific purpose. This process ensures that all critical parameters are known and determines the influence of various factors on the obtained results. The elements of validation that can be obtained during its performance are mainly linearity, precision, accuracy, limit of detection (LOD) and limit of quantification (LOQ). These values allow the capacity to determine the credibility and limitations of the developed method, which is especially important in the quantitative analysis [43].

#### 2.7.1. Specificity

Specificity ensures the unambiguous determination of the test substance, free from the influence of impurities and auxiliary substances that may be present in the probe. In addition, a properly established analytical procedure makes it possible to distinguish compounds of similar structure by comparing the results of the analysis of the test probe with the reference.

#### 2.7.2. Linearity Range

The linearity of an analytical method means that in a given concentration range, results and the recorded chromatographic peak area values are directly proportional to the concentration of the analyte in the sample (the quantitative analysis). In order to determine

the linearity, it is necessary to make measurements two to three times for at least five concentration levels (in the range of 50–150%), and then to determine the calibration curve, which can be described by the following equation:

$$y = ax + b$$

where y—peak area of the chromatographic peak [mm$^2$], x—concentration of the analyte in the sample, a—coefficient of the slope of the calibration curve and b—coefficient of the intercept of the calibration curve with the y axis.

For this purpose, standard solutions of tested B vitamins with concentrations from 5 to 300 µg/mL (for B1, B2, B6 and B12) and 50 to 600 µg/mL (for B5) have been prepared and applied to the plates in a volume of 10 µL.

### 2.7.3. Limit of Detection (LOD) and Limit of Quantification (LOQ)

The limit of detection (LOD) is the smallest amount (concentration) of an analyte in a sample that can be detected. This limit is greater than the upper limit of the noise level in a given measurement process. In order to determine it, one of the following methods should be used, listed below:

- Visual assessment: analysis of samples with known analyte concentrations and establishment of a minimum level at which the analyte can be reliably detected;
- Determining the ratio of the analytical signal to the average background noise level of the blank sample. This can be completed by comparing the signal sizes of samples with a known low analyte content with the values obtained for blank samples;
- Determination from the parameters of the calibration curve made in low concentration ranges, using the following formula:

$$LOD = \frac{3.3 \times Sb}{a}$$

The limit of quantification (LOQ) is the smallest amount (concentration) of an analyte in a sample that can be quantified with adequate accuracy, precision and accuracy. LOQ can be determined similarly to LOD: visually, from the ratio of the analyte signal to noise or from the parameters of the calibration curve, it is calculated from the formula:

$$LOQ = \frac{10 \times Sb}{a}$$

where Sb—standard deviation of the intercept, *a*—slope of the calibration curve.

### 2.7.4. Precision

Precision is a parameter that determines how consistent the results obtained by repeating the repeated determinations of one sample with a given method. With the increase in precision, the dispersion of individual results in relation to the average obtained from them decreases. This parameter depends on random errors, i.e., small errors caused by disturbances during the measurement. The term "precision" covers the following terms: repeatability, reproducibility and intermediate (inter-day) precision. Repeatability (intra-day) refers to analyses carried out in a short time by the same method, in one laboratory, by one person, using the same instruments and reagents. Intermediate precision is in turn related to the analysis of the same sample in the same laboratory but with different devices by different researchers. It allows the capacity to determine the influence of personal and equipment factors on the measurement values. Reproducibility refers to tests carried out with the same method and sample by different analysts in different laboratories with different equipment. To determine precision, at least six replicate results at 100% or nine results within the range of the method (e.g., three replicates at three different concentrations) must be made. Precision can be measured by standard deviation (SD), relative standard deviation RSD (independent of measurement units) and coefficient of variation CV (RSD

expressed as a percentage). The CV for the main component of the sample should be up to 2–3%.

### 2.7.5. Accuracy

Accuracy allows the capacity to determine the compliance of the obtained result of a single measurement with the real value. There are several ways to determine accuracy: analyzing a sample containing an exact known amount of analyte and comparing the results with the true value; comparing the obtained results with the results received by another method; and determining the percentage recovery of the analyte added to the test sample. The recovery for the analyte should be in the range of 95–105%. Determination of the accuracy requires a minimum of nine determinations involving at least three different concentration levels (most often 80%, 100%, 120%) in a given linearity range.

### 2.7.6. Robustness

The robustness parameter makes it possible to list actions whose modification may affect the final result of the analysis. To ensure that slight fluctuations in temperature, extraction time, pH or mobile phase composition do not compromise the reliability of the assay, the analytical conditions to which the measurements are subjected should be controlled.

### *2.8. Statistical Analyses*

The analysis was carried out using Statistica v.13.3. TIBCO Software Inc (Palo Alto, CA, USA). The confidence limit of $p < 0.05$ was considered as statistically significant.

### 3. Results

One of the chromatographic techniques, using physicochemical phenomena to separate the components of the mixture and enabling their simultaneous qualitative and quantitative assessment, is TLC, especially coupled with densitometric detection. It is commonly used in experimental research in many medical studies, in particular due to particular significant advantages over most common HPLC or GC techniques [44]. In the presented work, we used the TLC with densitometric detection to develop the conditions for the determination of B vitamins. In order to obtain the optimal conditions for the separation of the mixture of five vitamins, the various types of chromatographic plates were tested. Moreover, to optimize the conditions for the determination of vitamin B5 TLC Cellulose F stationary phases were checked, temperature activated (drying the plate in an oven for 24 h at 60 °C) and activated with 1% β-cyclodextrin solution: methanol and then drying in an oven for 24 h at 60 °C. Furthermore, different mobile phases were experimentally tested with the composition of mixtures of different solutions. For the determination of vitamin B5, two additional mobile phases were tested as follows: n-hexane: ethyl acetate (9:1 *v/v*) and 1% solution of β-cyclodextrin: methanol (15:1 *v/v).* After developing in tested phases, the chromatographic plates were assessed visually under a UV lamp at wavelengths 254 and 366 nm. Vitamin B12, red in a color (Figure 2), was also visible in daylight, while vitamin B5 turned purple after staining with 1% ninhydrin solution.

Comparing the results for all types of plates ($R_F$ values, separation of individual peaks corresponding to individual vitamins and their shape), a good separation of five tested vitamins was obtained for TLC Silica gel 60 $F_{254}$ plates and the mobile phase composed of the following: chloroform: ethanol: water: glacial acetic acid (2:8:2:0.5 *v/v/v/v).* The mobile phase containing ammonia was not suitable for the intended determination, as the ammonia contained therein reacts with ninhydrin. The selected analysis conditions allowed us to obtain compact spots on the chromatograms and a good peak shape on the registered densitograms. Additionally, the absorption spectra of the tested vitamins in the range from 200 to 400 nm were recorded. Based on these observations, wavelengths of 254 nm (for vitamins B1, B2, B6, B 12) (Figure 3), 343 nm (for vitamin B12) and 550 nm (for vitamin B5 stained with ninhydrin) were selected for drug quantification. The retardation

factors ($R_F$) determined for the vitamin B in the discussed conditions are B1 0.18, B2 0.81, B5 0.42, B6 0.75 and B12 0.33.

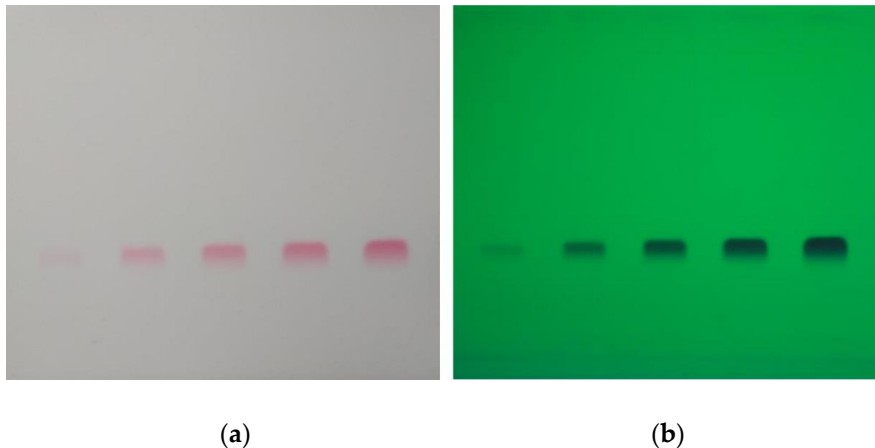

(a)                                                                          (b)

**Figure 2.** An example of chromatograms (image) for vitamin B12 with different concentrations (in the linearity range) registered in (**a**) daylight and (**b**) under a UV lamp (254 nm).

Additionally, different developing lengths (95 and 145 mm) of tested stationary phase were checked. A similar distribution and recorded $R_F$ values were obtained, and finally the plates were developed to 95 mm.

The recorded values of the $R_F$ and the absorption spectra for each B vitamin were used for the qualitative analysis, and the chromatographic peak area values for the quantitative analysis.

In the next step of our work, the optimized method was validated in accordance with ICH requirements in terms of its reliability [43]. The specificity of the method was assessed for possible regrowth contaminations and interference from the auxiliary in pharmaceutical products. The registered chromatograms and absorption spectra of all vitamin solutions obtained from pharmaceutical products did not exhibit any supplementary peaks compared to those obtained for standard substances. It can be concluded that the proposed procedure is selective and specific for the analyzed vitamins.

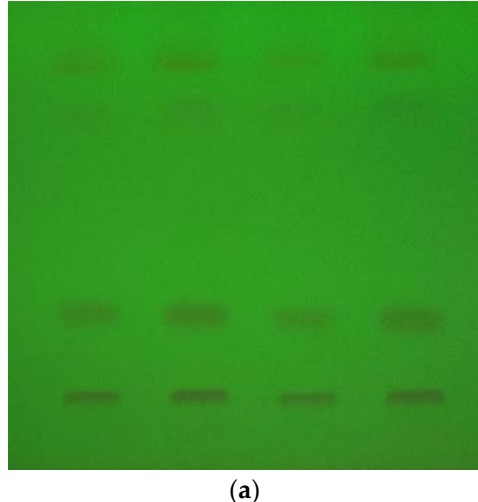

(a)

**Figure 3.** *Cont.*

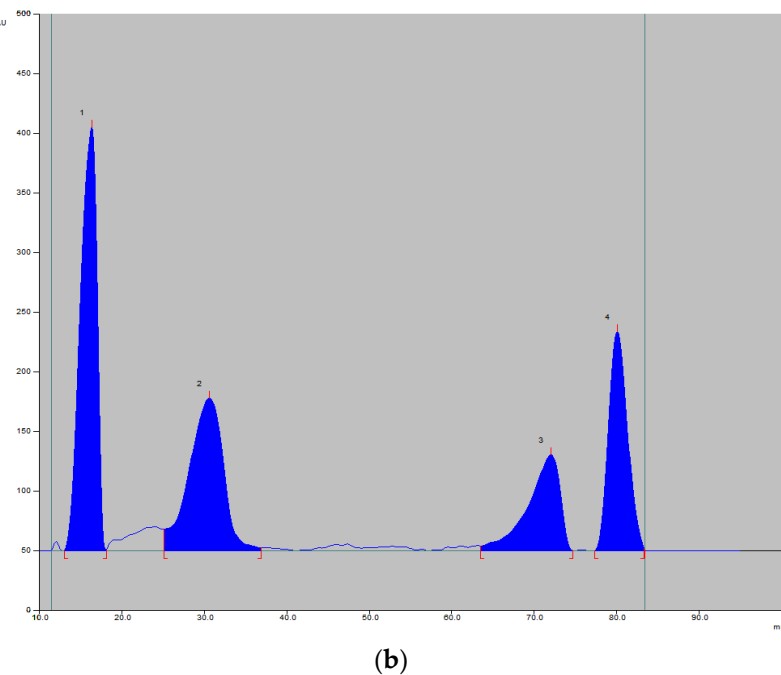

(**b**)

**Figure 3.** An example of a chromatogram under a UV lamp (254 nm; image) (**a**) and registered densitogram (**b**) showing the separation of vitamins 1—B1 ($R_F$ = 0.18), 2—B12 ($R_F$ = 0.33), 3—B6 ($R_F$ = 0.75), 4—B2 ($R_F$ = 0.81).

Next, the validation of the optimized procedure in terms of the linearity interval was determined. After developing in the chosen mobile phase, in the case of the B5 vitamin, stanning with ninhydrin and densitometric detection were performed, and peak areas corresponding to a specified concentration were registered. The obtained calibration curves confirmed a very good fit of the regression line to the real data (Figure 4).

The confidence interval around the regression lines contain each point, which confirms the correlation between the tested variables. The correlation coefficients (r) (in all linearity ranges for B1 in the range from 10 to 170, B2 10 to 250, B5 100 to 500, B6 30 to 200 and B12 10 to 170 µg/mL) show the stronger correlation relationship, and were close to 1. In addition, other statistical parameters such as the standard deviation of the slope, the standard deviation of the intercept and the standard error of the estimate have smaller values, confirming that the model is well matched. Based on the obtained data, the limits of detection (LOD) and quantification (LOQ) were calculated (Table 1). Obtained parameters were characterized by relatively low values, which indicates that the developed method is sufficiently sensitive to the analyzed vitamins, and confirms the high predictive ability of the developed method.

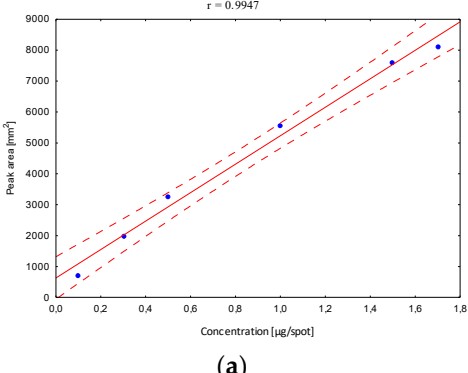

(**a**)

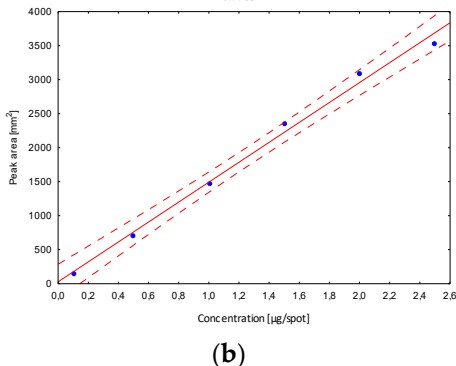

(**b**)

**Figure 4.** *Cont.*

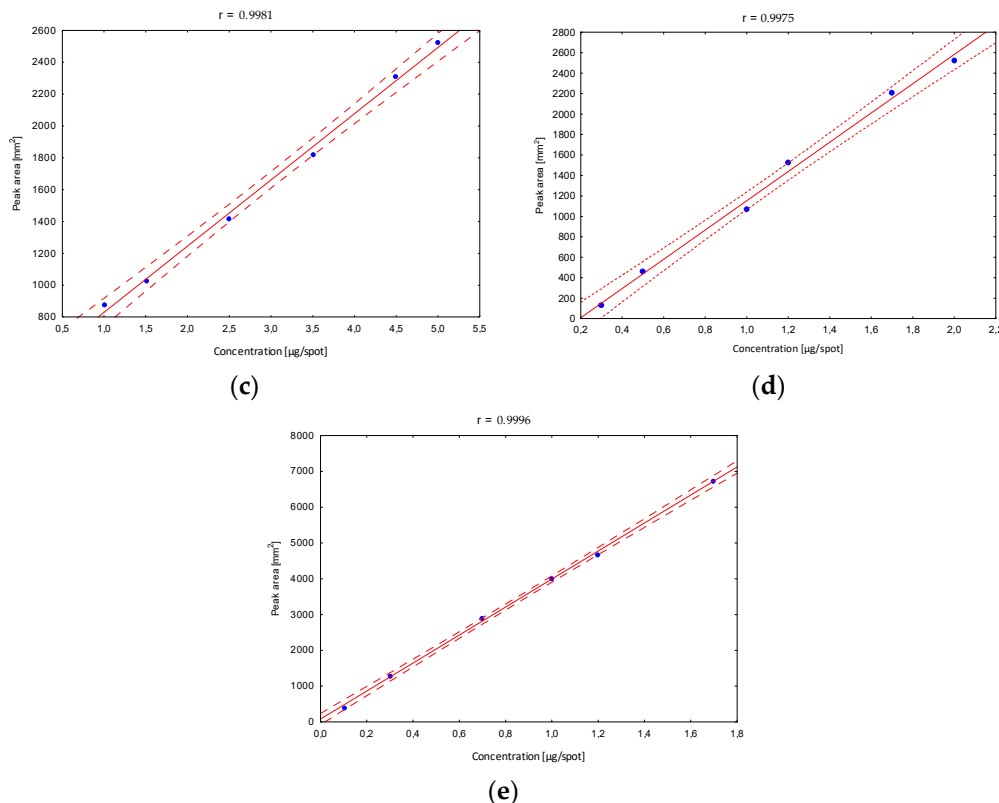

**Figure 4.** The linearity range of determined B vitamins; (**a**) B1, (**b**) B2, (**c**) B5, (**d**) B6, (**e**) B12.

**Table 1.** The linearity, LOD and LOQ results of method validation.

| Vitamin | Linearity Parameters | LOD [μg/Spot] | LOQ [μg/Spot] |
|---------|---------------------|---------------|---------------|
| B1 | a = 4605.06; b = 237.93<br>Sa = 629.50; Sb = 247.46<br>r = 0.9947; r2 = 0.9894<br>Se = 349.28 | 0.1773 | 0.5374 |
| B2 | a = 1465.10; b = 61.75<br>Sa = 27.86; Sb = 93.51<br>r = 0.9965; r2 = 0.9929<br>Se = 125.54 | 0.2106 | 0.6382 |
| B5 | a = 416.12; b = 413.19<br>Sa = 12.79; Sb = 42.74<br>r = 0.9981; r2 = 0.9962<br>Se = 46.11 | 0.3389 | 1.0271 |
| B6 | a = 1429.99; b = 50.40<br>Sa = −277.90; Sb = 63.98<br>r = 0.9975; r2 = 0.9951<br>Se = 74.55 | 0.1615 | 0.4474 |
| B12 | a = 3912.43; b = 56.46<br>Sa = 79.77; Sb = 56.08<br>r = 0.9996; r2 = 0.9992<br>Se = 74.76 | 0.0473 | 0.1433 |

where a—the slope of calibration curve, b—the intercept, Sa—standard deviation of the slope of the calibration curve, Sb—the standard deviation of the intercept, r—regression coefficient, Se—standard error of estimation.

The precision was determined by applying 10 μL of standard solutions for each vitamin to the plates, and then carrying out the chromatographic separation and densitometric

detection as described above. The precision was assessed by analyzing peak areas registered for the tested vitamins on the same day (intra-day) and after 7 days (inter-day precision). Each analysis was performed three times. The registered peak area values were analyzed statistically, and it can be concluded that the received RSD coefficients confirmed the method to be considered precise. The obtained results are summarized in Table 2.

**Table 2.** The results of the inter- and intra-day precision of the determination of B vitamins with statistical evaluation.

| Vitamin | Intra-Day Precision | | Inter-Day Precision | |
|---|---|---|---|---|
| | Peak area [mm$^2$], $n = 3$ | Statistical parameters | Peak area [mm$^2$], $n = 3$ | Statistical parameters |
| B1 | $\bar{x} = 5294.00$ | $S\bar{x} = 29.82$ <br> SD = 66.69 <br> $\mu = 5294.00 \pm 191.63$ <br> RSD = 1.26% | $\bar{x} = 5399.46$ | $S\bar{x} = 29.03$ <br> SD = 64.90 <br> $\mu = 5399.46 \pm 186.50$ <br> RSD = 1.2% |
| B2 | $\bar{x} = 2060.30$ | $S\bar{x} = 7.09$ <br> SD = 15.86 <br> $\mu = 2060.30 \pm 45.58$ <br> RSD = 0.77% | $\bar{x} = 2155.08$ | $S\bar{x} = 12.71$ <br> SD = 28.44 <br> $\mu = 2155.08 \pm 81.71$ <br> RSD = 1.32% |
| B5 | $\bar{x} = 4848.88$ | $S\bar{x} = 32.94$ <br> SD = 73.66 <br> $\mu = 4848.88 \pm 211.67$ <br> RSD = 1.52% | $\bar{x} = 4972.80$ | $S\bar{x} = 31.19$ <br> SD = 69.74 <br> $\mu = 4972.80 \pm 200.41$ <br> RSD = 1.40% |
| B6 | $\bar{x} = 2060.60$ | $S\bar{x} = 11.82$ <br> SD = 26.44 <br> $\mu = 2060.60 \pm 75.9785$ <br> RSD = 1.28% | $\bar{x} = 2153.02$ | $S\bar{x} = 11.15$ <br> SD = 24.94 <br> $\mu = 2153.02 \pm 71.66$ <br> RSD = 1.16% |
| B12 | $\bar{x} = 4160,98$ | $S\bar{x} = 11.51$ <br> SD = 25.73 <br> $\mu = 4160.98 \pm 73.95$ <br> RSD = 0.62% | $\bar{x} = 4272.10$ | $S\bar{x} = 16.14$ <br> SD = 36.09 <br> $\mu = 4272.10 \pm 103.71$ <br> RSD = 0.84% |

where $\bar{x}$—arithmetic average, $S\bar{x}$—standard deviation (standard error) of the arithmetic mean, SD—standard deviation, $\mu$—confidence interval of the arithmetic mean with a probability of 95%, RSD—relative standard deviation [%].

The accuracy of the established method was experimentally determined as the percent recovery of a known amount of analyte added to the tested sample. The determinations were carried out at three concentration levels. For this purpose, standard solutions of testing vitamins B and supplements containing appropriate the vitamin were prepared and then applied on the stationary phase in the 10 mm bands (standard solutions, sample solutions and a mixture containing 80, 100 and 120% of standard solution of appropriate vitamin added to the supplements). The calculated results show the good accuracy of the method (ranging from 98.73 to 100.96% with the RSD (in %) from 0.37 to 0.70 (Table 3)). The robustness of the validated procedure was conducted by the reliability of the analysis with respect variations in the experimental conditions, e.g., the chamber size ($27 \times 27 \times 7$ cm, $12 \times 11 \times 8$ cm), chamber saturation time ($\pm 5$ min), the development distance ($\pm 0.5$ cm) and the volume of individual components in the selected mobile phase ($\pm 0.1$ mL). Based on the obtained $R_F$ values and absorption spectra, no significant differences in chromatographic behavior were found for all analyzed vitamins, which indicates the robustness of the method.

**Table 3.** The results of the accuracy of the determination of B vitamins with statistical evaluation.

| Vitamin | Recovery Level 80%, *n* = 3 | Recovery Level 100%, *n* = 3 | Recovery Level 120%, *n* = 3 |
|---|---|---|---|
| B1 | $\bar{x}$ = 99.81<br>S$\bar{x}$ = 0.23; SD = 0.46<br>μ = 99.81 ± 1.70<br>RSD = 0.46% | $\bar{x}$ = 100.60<br>S$\bar{x}$ = 0.22; SD = 0.44<br>μ = 100.60 ± 1.65<br>RSD = 0.44% | $\bar{x}$ = 100.96<br>S$\bar{x}$ = 0.19; SD = 0.38<br>μ = 100.96 ± 1.43<br>RSD = 0,37% |
| B2 | $\bar{x}$ = 98.90<br>S$\bar{x}$ = 0.17; SD = 0.33<br>μ = 98.90 ± 1.23<br>RSD = 0.33% | $\bar{x}$ = 100.15<br>S$\bar{x}$ = 0.28; SD = 0.56<br>μ = 100.15 ± 2.07<br>RSD = 0.55% | $\bar{x}$ = 99.75<br>S$\bar{x}$ = 0.19; SD = 0.38<br>μ = 99.75 ± 1.40<br>RSD = 0.38% |
| B5 | $\bar{x}$ = 100.16<br>S$\bar{x}$ = 0.21; SD = 0.42<br>μ = 100.16 ± 1.56<br>RSD = 0.41% | $\bar{x}$ = 99.95<br>S$\bar{x}$ = 0.32; SD = 0.63<br>μ = 99.95 ± 2.36<br>RSD = 0.63% | $\bar{x}$ = 99.89<br>S$\bar{x}$ = 0.26; SD = 0.53<br>μ = 99.89 ± 1.97<br>RSD = 0.53% |
| B6 | $\bar{x}$ = 99.61<br>S$\bar{x}$ = 0.23; SD = 0.46<br>μ = 99.61 ± 1.71<br>RSD = 0.46% | $\bar{x}$ = 99.17<br>S$\bar{x}$ = 0.11; SD = 0.22<br>μ = 99.17 ± 0.81<br>RSD = 0.22% | $\bar{x}$ = 99.01<br>S$\bar{x}$ = 0.27; SD = 0.55<br>μ = 99.01 ± 2.04<br>RSD = 0.56% |
| B12 | $\bar{x}$ = 98.73<br>S$\bar{x}$ = 0.34; SD = 0.69<br>μ = 98.73 ± 2.56<br>RSD = 0.70% | $\bar{x}$ = 100.09<br>S$\bar{x}$ = 0.29; SD = 0.58<br>μ = 100.09 ± 2.15<br>RSD = 0.58% | $\bar{x}$ = 99.71<br>S$\bar{x}$ = 0.25; SD = 0.49<br>μ = 99.71 ± 1.83<br>RSD = 0.49% |

where $\bar{x}$—arithmetic average, S$\bar{x}$—standard deviation (standard error) of the arithmetic mean, SD—standard deviation, μ—confidence interval of the arithmetic mean with a probability of 95%, RSD—relative standard deviation [%].

Analyzing the parameters obtained during the validation, such as specificity, linearity, limits of detection and quantification, precision, and accuracy, it can be concluded that the developed methods meet all the criteria required for an analytical procedure intended for the qualitative and quantitative control of medicinal products.

In the next step, using the validated method, nine selected medicinal products and dietary supplements containing single or several vitamins were analyzed. The obtained values of the areas of the chromatographic peaks were used to determine the content of particular vitamin B in the tablet or capsule. The results of TLC analysis coupled with densitometric detection and their statistical evaluation are presented in Table 4. The obtained results were in most cases consistent with the manufacturer's declaration for tested preparations. In turn, in the case of Product E, a deficiency of the active substance (vitamin B12) was detected, and the marked content was only 12% of the quantity declared by the manufacturer. Similarly, though smaller, differences were registered between the determined and declared content of vitamins in the case of the analysis of Product F (vitamin B6) and Product G (vitamins B5 and B6).

**Table 4.** The determined content of B vitamins in the analyzed preparations.

| Preparation | | Declared Content | Determined Content (*n* = 3) | Statistical Evaluation (*n* = 3) |
|---|---|---|---|---|
| Product A | B6 | 50 mg/tablet | 50.04 mg/tablet | S$\bar{x}$ = 0.26; SD = 0.59<br>μ = 50.04 ± 1.70<br>RSD = 1.18% |
| Product B | B6 | 50 mg/tablet | 51.17 mg/tablet | S$\bar{x}$ = 0.36; SD = 0.81<br>μ = 51.17 ± 1.59<br>RSD = 1.59% |

**Table 4.** *Cont.*

| Preparation | | Declared Content | Determined Content (*n* = 3) | Statistical Evaluation (*n* = 3) |
|---|---|---|---|---|
| Product C | B1 | 3 mg/tablet | 0.34 mg/tablet | S$\overline{x}$ = 0.002; SD = 0.006<br>μ = 0.34 ± 0.02<br>RSD = 1.6% |
| Product D | B1 | 35 mg/tablet | 34.78 mg/tablet | S$\overline{x}$ = 0.27; SD = 0.60<br>μ = 34.78 ± 1.72<br>RSD = 1.72% |
| Product E | B12 | 0.7 mg/tablet | 0.0168 mg/tablet | S$\overline{x}$ = 0.0002; SD = 0.0004<br>μ = 0.0168 ± 0.0013<br>RSD = 2.38% |
| Product H | B2 | 3 mg/tablet | 3.07 mg/tablet | S$\overline{x}$ = 0.02; SD = 0.04<br>μ = 3.07 ± 0.12<br>RSD = 0.6% |
| Product F | B1 | 2.4 mg/tablet | 2.56 mg/tablet | S$\overline{x}$ = 0.02; SD = 0.05<br>μ = 2.56 ± 0.14<br>RSD = 1.68% |
| | B6 | 4.1 mg/tablet | 3.73 mg/tablet | S$\overline{x}$ = 0.01; SD = 0.03<br>μ = 3.73 ± 0.08<br>RSD = 1.05% |
| | B5 | 1.8 mg/tablet | 1.75 mg/tablet | S$\overline{x}$ = 0.04; SD = 0.08<br>μ = 1.75 ± 0.08<br>RSD = 4.28% |
| Product G | B1 | 3 mg/tablet | 2.71 mg/tablet | S$\overline{x}$ = 0.01; SD = 0.02<br>μ = 2.71 ± 0.06<br>RSD = 0,76% |
| | B6 | 5 mg/tablet | 3.22 mg/tablet | S$\overline{x}$ = 0.02; SD = 0.04<br>μ = 3.22 ± 0.11<br>RSD = 1.2% |
| | B5 | 5 mg/tablet | 4.00 mg/tablet | S$\overline{x}$ = 0.01; SD = 0.06<br>μ = 4.00 ± 0.10<br>RSD = 2.5% |
| Product I | B5 | 500 mg/caps | 500.71 mg/caps | S$\overline{x}$ = 0.24; SD = 0.53<br>μ = 500.71 ± 1.52<br>RSD = 0.11% |

where $\overline{x}$—arithmetic average, S$\overline{x}$—standard deviation (standard error) of the arithmetic mean, SD—standard deviation, μ—confidence interval of the arithmetic mean with a probability of 95%, RSD—relative standard deviation [%].

Summing up, the obtained results prove the possibility of using the developed method for the quantification of selected B vitamins in pharmaceutical products and dietary supplements with appropriate precision, accuracy and sensitivity.

## 4. Conclusions

In the presented paper, a method of the pharmaceutical determination of vitamins B1, B2, B5, B6 and B12 in pharmaceutical preparations based on in thin layer chromatography with densitometric detection was developed, optimized and validated according to the ICH guidelines. Under the described conditions, the complete separation of the five tested vitamins was observed.

The conducted analysis showed that none of the excipients present in the vitamin preparations interfered with both qualitative and quantitative determination. The developed procedure enables the simultaneous determination of five vitamins during one analytical procedure, which was confirmed by obtaining a good separation of separated peaks (derived from individual vitamins) on the chromatograms. The proposed method of simultaneous determination of vitamins B1, B2, B5, B6 and B12 is very simple, cheap and

fast, and gives precise and accurate results. The proposed procedure can be a valuable tool in quality control laboratories and an alternative to more advanced methods for determining the content of B vitamins in both drugs and commonly used dietary supplements.

**Author Contributions:** Conceptualization, M.D. and J.Ż.; methodology, M.D. and J.Ż.; software, J. Ż.; formal analysis, Ż.B.-K. and M.S.; investigation, J.Ż. and Ż.B.-K.; writing—original draft preparation, J.Ż.; writing—review and editing, M.D. and M.S.; supervision, M.D. All authors have read and agreed to the published version of the manuscript.

**Funding:** This research received no external funding.

**Data Availability Statement:** Not applicable.

**Conflicts of Interest:** The authors declare no conflict of interest.

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
