# Peer review of "Development and Optimization of Chromatographic Conditions for the Determination of Selected B Vitamins in Pharmaceutical Products"

_processes, doi:10.3390/pr11030937_

Round 1

Reviewer 1 Report

Comments

The authors determine in this article the best conditions of the chromatographic method for the detection of vitamins B1, B2, B5, B6, and B12. Although the article is interesting, I have some comments that I think can improve the document.

Abstract

Line 18 Typo …  v/v /v/v)

There is a space that is not necessary.

Introduction 

Figure 1, chemical structures quality must be improved.

Line 60, There is a double space between “of” and “water”.

The determination of  water-soluble vitamins is quite dif

Line 61, typo. “time-consuming”

Sometimes they use “Thiamin” (line 67) and sometimes Thiamine (lines 74, 76, 78, etc.). Homogenize.

Although the introduction extensively describes the different methods used to determine vitamins B1, B2, B5, B6, B12, the introduction is very extensive. According to Instructions for Author from Processes “The introduction should briefly place the study in a broad context and highlight why it is important. It should define the purpose of the work and its significance, including specific hypotheses being tested. The current state of the research field should be reviewed carefully and key publications cited. Please highlight controversial and diverging hypotheses when necessary. Finally, briefly mention the main aim of the work and highlight the main conclusions. Keep the introduction comprehensible to scientists working outside the topic of the paper.

Materials and Methods

Line 223, were purchased from Sigma-Aldrich (USA).

Line 224, what is the meaning of OTC, please define it.

Line 234, 1% β-cyclodextrin water solution

Line 235, 1% ninhydrin etanolic solution

Line 244, a space between 20 and ×20 “Chromatographic plates in size 20 ×20 cm

Line 245, Typo, Type E.

What is the purpose of mentioning the CAS registry number of the plates?

Sometimes number is abbreviated as "No" and sometimes "No." Homogenize.

Review the format of the subtitles, the first letter of the words must be capitalized, e.g.

Line 249, 2.4. Standard and Sample Solutions

Line 256, 2.5. Optimization of Determination Conditions

Please correct the following:

Line 262, 1% β-cyclodextrin : methanol (15:1 v/v).

Line 266, with 1% ninhydrin ethanolic solution

Line 267, use the hour symbol (h) “the chromatograms were allowed to dry for 24 h”

Line 268, 1% ninhydrin ethanolic solution

Line 270, These should be mentioned in the Results section:

“Comparing the areas of the peaks, it was found that the elongation heating time does not have a significant effect on the results.”

Line 272, Use symbols for your units: “10 min at a temperature of 85 °C” (space between the number and the degrees Celsius).

Line 273, Use symbols for your units: “approx. 10 min (at room temperature)”

Line 289, space between the number and the degrees Celsius. “an oven (10 min, at 85 °C), densitometically

Lines 291-292, “and the recorded chromatographic peak area values for the quantitative analysis.” and section 2.7.2.

Were calibration curves made for the quantitative analysis? If so, at what concentrations were the standards used for the calibration curves? What were the equations of the line obtained?

Line 293, Method Validation

Subtopics 2.7.1-2.7.6 do not have citations, it would then be understood that their corresponding citations are [68,69], is that correct?

Line 374, Please add the country of the software provider.

Line 391 1% β-cyclodextrin solution: methanol.

Figure 2 does not indicate the retardation factors, nor the start or end guide lines that the winCATS software provides, and the concentrations of vitamin B1 used are not indicated in figure 2 either.

Figure 3 (a) does not indicate the retardation factors, nor the start or end guide lines that the winCATS software provides, nor does it indicate which vitamins are B1, B2, B6, or B12.

Figure 3 (b) does not clearly distinguish the names of the X and Y axes, nor their units, nor does it distinguish which is vitamin B1, B2, B6, or B12.

Lines 430-432, this information is requested in the methodology section.

Figure 4, part d) should be e). The correlation coefficients (r) are not indicated in the figure, indicate.

Tables 3 and 4 do not indicate the footnote to the table where the meaning of the acronyms is indicated.

Author Response

-

Reviewer 2 Report

The present study proposed the method for the development and optimization of chromatographic conditions for the determination of selected B vitamins in pharmaceutical products. The work is interesting, but some of the descriptions in this manuscript were very confusing. Therefore, I recommend it can be published in processes after major revisions as follows.

(1)    In line 48, As far as I know, there should be a lot of literature on the use of techniques in vitamin-related research, and authors should cite recent relevant work.

(2)    What are the advantages of the approach proposed in this manuscript? The author should highlight this advantage.

(3)    The quantitative results obtained by the proposed method should be verified using conventional methods, such as liquid chromatography, to ensure the effectiveness of the proposed method.

(4)    The section 2.2 should be rewrited.

(5)    In line 90-92, the sentence of Vitamin B1 is a cofactor for many enzymes whose 90 percentage increase in activity is used can be used to determine the thiamine content 91 in a sample. should be rewrited.

(6)    Figure 4 has 5 subplots, please indicate what these 5 subplots represent.

 The authors should check the whole manuscript carefully and make the manuscript more readable.

Author Response

-

Reviewer 3 Report

the research framework is very interesting, and the paper is well written and organized with clear focus on the all the aspects of the aim of the study. the TLC determination of vitamins is rapid and low-cost analytical techniques. in addition, it can be also reliable if the method is proper validated. the authors successfully respond to all the parameters of method validation, but I have few suggestions:

I- the authors said that they tried various stationary phase and they selected the silica gel it would be good if authors add paragraph about what was the criteria for selecting the silicagel as most suitable stationary phase (for example was resolution of analytes, or was it composition of mobile phase, or peak shapes ......) 

II- Are the determined limits of detection and quantification in accordance with ICH guidance I don't get that impression when I compare it?

III- Could the authors explain why they use the linear regression when the scatter plot suggests exponential function. maybe the better selection of the linear range could improve the scatter plots.

IV- the references count shows 71 reference which is in my opinion too much and makes confusion to reader i suggest authors to improve selection of references.

Author Response

-

Round 2

Reviewer 2 Report

The author has revised the article according to the suggestions. I think this manuscript can be published in Processes.

Reviewer 3 Report

after implemented improvements the article looks much better and fulfil the minimum requirements for publication.